# Change in Urban Land Use Efficiency in China: Does the High-Speed Rail Make a Difference?

**DOI:** 10.3390/ijerph181910043

**Published:** 2021-09-24

**Authors:** Wenyi Qiao, Xianjin Huang

**Affiliations:** School of Geographic and Oceanographic Sciences, Nanjing University, Nanjing 210023, China; qiaowenyidl@163.com

**Keywords:** high-speed railway, urban land use efficiency, difference-in-difference, threshold effect

## Abstract

High-speed rail (HSR) increases the non-local connections in cities and plays an essential role in urban land use efficiency. This paper uses a multi-period difference-in-difference model and a threshold model based on sample data that cover 284 Chinese cities from 2003–2018 to investigate the impact of HSR on urban land use efficiency. The results show that there is a 0.021 increase in urban land use efficiency after opening the HSR. The number of HSR stations and routes can increase the urban land use efficiency by 0.004 and 0.013, respectively. Compared with the cities in the East, the midwestern ones are more vulnerable to the impact of HSR. In particular, the positive impact of the number of HSR stations on the urban land use efficiency in cities with an urban population density exceeding 795 person/km^2^ is two times larger than cities with an urban population density of less than 795 person/km^2^. In addition, the impact of the number of HSR routes on urban land use efficiency in cities with an urban population density of less than 1003 person/km^2^ is five times larger than that of cities with an urban population density exceeding 1003 person/km^2^.

## 1. Introduction

The rapid development of urbanization makes cities face huge challenges, such as how to keep urban growth under limited land supply and resource constraints. In this regard, the straightforward idea is to raise the urban land use efficiency (ULUE) [1]. ULUE refers to the maximum economic benefits attainable in the output of a unit land area [2]. As an open system, the improvement of ULUE isn’t just a purely urban endogenous process that is more dependent on the non-local connections. Non-local connections represent the exchange process of elements between one city and other cities [3], reflecting the resource reallocation that conforms to the spatial division of labor [4], thereby greatly improving the input and output efficiency of urban land use. In order to further strengthen resource reallocation between regions, China has formulated a grand strategy for a national high-speed rail (HSR) system connecting 250 cities with a total mileage of 45,000 km by 2030. This process indicates that the construction of HSR to enhance the non-local connections of cities has become an unstoppable trend [5,6]. Hence, in-depth discussion the impact and mechanisms of HSR on ULUE has become an important concern.

Developing countries are enthusiastic about building modern HSR, owing the significant space-time compression and scale effect of HSR [7]. Cities with HSR may carry out a large amount of land development near the stations [8] (such as HSR new towns). Policy makers believe that the construction of a large-scale transportation infrastructure will help the diffusion of capital, talents, technology, and other production factors from central cities to peripheral cities, promote the industrialization and economic growth of cities along the HSR lines, and balance regional development.

A substantial proportion of the literature has discussed the impact of HSR on urban land use, but the views are not consistent. For example, although accessibility has improved in China from 2008–2013 due to the large-scale development of HSR [9], scholars found that neither the HSR in China nor that in the EU have significantly changed the spatial pattern of urban land use [4]. Conversely, some research concluded that the increasing benefits from HSR have spread to regions along the routes, becoming a powerful driving force for local urban land expansion [10]. For example, Cui et al. took 42 cities in the Shandong Peninsula urban agglomerations as an example and stated that HSR can reconstruct the spatial pattern of urban land use [11]. Hessen (2013) further pointed out that the extent of the HSR network may have mixed effects on land use between regions [12]. In addition, some scholars believe that this influence has regional heterogeneity [13].

Although plenty of research has explored the efficiency of HSR, the knowledge of the impact of HSR on ULUE is still limited [14]. The commonality of previous studies was that the city was regarded as an isolated system, then dummy variables were evaluated for time and regional differences in the impact of HSR [4] while ignoring the important force behind the ULUE (such as non-local connections). In addition, the impact of HSR stations and routes was often ignored in land use. More importantly, considering that the improved accessibility and the increase in non-local connections brought about by HSR occurs gradually [15], it is also unclear whether the HSR has any “threshold effect”.

This paper aims to evaluate the ULUE via a data envelopment analysis model (DEA), examine the impact of HSR on ULUE, and analyze the regional heterogeneity and threshold effect of the ULUE. The likely significance of this paper is twofold. First, the improvement of ULUE is conducive to the urban sustainable development. Second, the large-scale construction of HSR connecting many cities in China is overwhelming. Therefore, it is important to examine the impact of HSR on ULUE in different regions, as this can provide valuable implications for future HSR development and land planning in turn.

The innovation of this paper has three aspects. First, this paper re-examined the impact of HSR on ULUE from the perspective of non-local connections. Second, a multi-phase difference-in-difference (DID) method was utilized to investigate whether the opening of HSR, the number of HSR stations and routes had an impact on ULUE. Finally, a threshold effect model was used to test whether HSR has a threshold effect.

The rest of this paper is organized as follows. Section 2 reviews the related literature and the theoretical expectations. Section 3 describes the research design. Section 4 presents the main results. Section 5 and Section 6 provide the discussion and conclusions, respectively.

## 2. Literature Review and Theoretical Expectations

### 2.1. Literature Review

ULUE refers to the maximum economic benefits attainable in the output of a unit land area [16], which affects the economic growth and the construction of the human settlement environment. Many scholars have studied the spatial patterns and regional differences of ULUE [17], the measurement systems, and the influence mechanisms [18]. In the context of sustainable development, ULUE has shifted from caring about economic output to considering undesirable output (such as wastewater, waste gas, and smoke [19,20,21]). Hence, the ULUE in this paper is measured under the constraint of undesirable output.

For the measurement methods, different methods have different advantages and disadvantages. Specifically, the parametric method is based on a production function, but it is difficult to determine the specific distribution of errors [22]. The non-parametric method (data envelopment analysis (DEA)) is based on “Pareto Optimum” to find the relative effective point of each production unit on the production frontier, but it is not conducive to analyzing the specific time series changes [23]. Considering the DEA has a powerful advantage in overcoming the subjectivity brought by the specific expression of the input-output relationship and the determination of the weight of each indicator [24], this paper uses DEA to quantitatively evaluate the ULUE.

Existing studies indicated that urban land use structure, local government intervention in the land market, and changes in urban economic development factors (including the level of urbanization, labor transfer, and public investment) all have profound impacts on the ULUE. For example, Luo supported the view that the economic development, the degree of opening up, and the fixed asset investment can significantly promote the ULUE in Chinese cities [25]. Gianmi et al. stated that the larger the city, the higher the ULUE [26]. Classical location theory indicate that traffic accessibility is an important determinant of land use. With the large-scale construction of China’s HSR, the accessibility between cities has increased significantly, which will affect the exchange of social and economic activities between cities and change the structure and function of urban land use. Therefore, the impact of the transportation infrastructure with HSR as an important feature on ULUE has gradually received attention.

Plenty of empirical studies have provided a wealth of theoretical support for analyzing the impact of HSR on ULUE. It can be summarized into two viewpoints. Proponents of the positive impact of HSR on ULUE believed that HSR changes the professionalization pattern of cities by improving accessibility and industrial connections, and then reallocating tradable resources among cities [27]. This will generate a replacement effect, reduce the need for new land acquisition, and improve ULUE. Second, a city with developed HSR attracts the inflow of various resources [28], which increase the intensity of urban land use and greatly improve the ULUE. Recent literature has proven that HSR plays an important role in promoting economic growth, emission reduction, international technology transfer, knowledge spillover, and talent flow [29], especially in China. However, opponents insist that the HSR has decreased the ULUE. Although the HSR has accelerated the flow of factors such as population or enterprise migration, it was a single project in the initial stage, which aggravated the imbalance in the spatial distribution of factors and had a substantial negative impact on the ULUE [30]. At the same time, agglomerated negative externalities will increase the burden on cities with HSR (such as inefficient expansion, energy excessive consumption, and congestion effects), thereby increasing regional pollution emissions and density [31,32].

There were also some studies using statistical data to empirically analyze the impact of HSR on urban land value, urban land expansion, and urban land use structure. For example, Long et al. (2018) found that the urban expansion rate of cities with HSR was 0.12–0.13 faster than that of cities without HSR on average from 2004 to 2013 based on night light images [13]. Chen et al. (2020) found that HSR has different effects on different types of land use and supply according to 1.5 million Chinese land transaction records of [33]. The results indicated that the share of land for industrial purposes, logistics, and storage has decreased, while the supply of commercial and residential land has increased. This process will generate more economic benefits from a unit land area. In fact, the huge potential of the commercial, service, and real estate industries surrounding the HSR urges local governments to increase the supply of commercial and residential land in pursuit of the “HSR dividend”. Huang et al. (2021) found that HSR has a capitalization effect on real estate value based on China’s land transaction data [34]. This process will result accumulation of individual migration behavior and land finance, which will lead to low-density expansion of the entire urban space.

### 2.2. Theoretical Expectations

This paper investigates the impact mechanism of HSR on ULUE from scale efficiency (SE) and technological efficiency (TE).

The scale effect of HSR has two aspects. First, HSR links the originally isolated cities, which eliminates the obvious restrictions on administrative divisions [35,36,37]. This process will promote an increase in returns to scale and attract capital, a labor force, and other factors from other cities [38,39,40]. With the increase of factor inflow, the scale of land use investment continues to expand and the ULUE has been improved.

Another change is to promote the free flow of regional factors and reshape the distribution pattern of regional factors with the construction of HSR and the increase of routes and stations [4]. According to the new economic geography theory, factors have the characteristics of one-way flow from small cities, developing to big cities, or developed cities, leading to differences in the impact of ULUE in different cities [41]. For large cities or developed cities, the influx of factors will expand the scale of agglomeration and increase the ULUE. For small cities or developing cities, the reduction of production factors leads to the loss of ULUE.

On the other hand, excessive concentration of population and industry inevitably have a negative impact on the cities the factors flow into, causing pollution emission and energy consumption. Cities with more convenient HSR are always accompanied by excessive density of inputs. Hence, the congestion effect generated by agglomeration tends to exceed the economic effect of agglomeration, increasing the undesired outputs [42]. In other words, prominent problems such as external economics, traffic congestion, and environmental pollution will lead to a decline in ULUE.

There are different ways to improve the technical efficiency in various regions. Previous studies showed that the technical efficiency of cities with HSR mainly comes from the combination effect [43], which is mainly manifested through competition and learning effects. First, the agglomeration of factors has made local competition more and more fierce, forcing producers to upgrade technology and reduce costs, thereby increasing the ULUE [44]. Second, the agglomeration of elements has generated new knowledge through sharing and exchange effects. In particular, factors from the same or different industries gather in the same geographic space, creating an innovation environment for rapid knowledge transfer, thereby promoting the improvement of technical efficiency [45]. In addition, the increase in the labor force forms a specialized and skilled labor market, which reduces the cost of learning and improves the skills and production efficiency of workers, thereby raising the urban land technical efficiency.

However, it is worth noting that land capitalization inhibits technological progress. First, China’s land subsidy mechanism under the dual role of land finance and local economy will bring about the inefficient use of scarce land resources [46]. Second, the behavior of residents and companies investing large amounts of funds into the land market for the pursuit of high-profit investment preferences has reduced the investment in technology research and other fields.

There is the fact that the density and amounts of the HSR network present a spatial pattern of increasing from the West, the central region, and the East [47]. The economic development and urban growth in China exhibit the same spatial patterns, increasing from the poorer Western region to the richer Eastern region. The long-term existence of this difference has formed an urban system composed of cities at different stages of development. Therefore, there are reasons to believe that the impact of HSR on ULUE is bound to have regional heterogeneity. The impact of cities with different population sizes differs significantly, especially for megacities and cities with populations between 1–3 million [48]. In other words, the corridor benefits of HSR will only become prominent when the population density exceeds a certain threshold [49].

## 3. Research Design

### 3.1. Study Area and Data Source

This paper selects 284 prefecture-level cities and county-level cities from 2003 to 2018 in China. Given that some cities are seriously lacking in relevant socio-economic data, the cities of Sansha, Lasa, Tongren, Bijie, Pu’er, Longnan, Zhongwei, etc. are not included in this paper. The administrative scope of each prefecture-level city or county-level city is based on data from 2018.

This paper selects HSR with “C” and “G” prefixes as the research sample, including the opening time of HSR and the number of HSR stations and routes. Figure 1 shows the number and spatial distribution of cities that opened HSR in China from 2003 to 2018.

The data about HSR come from the official website of the China Railway Network. Land use data come from the Land Survey Change Data. Socio-economic data come from the China City Statistical Yearbook (2004–2019) and the China Statistical Yearbook (2004–2019). In this paper, there are three high-speed rail variables: whether to open high-speed rail (Wstation), the number of high-speed rail stations (Station), and the number of high-speed rail routes (Route). 

### 3.2. Variable Selection

#### 3.2.1. Explained Variables

Table 1 provides the indicators for measuring ULUE. Combining characteristics of urban land use and existing research results [50], the input indicators including all types of urban land, labor, and capital. Since the HSR system is land-consuming and occupies a non-negligible part of urban land, the impact in this paper actually refers to the HSR’s induced effects on ULUE rather than the efficiency of HSR’s own land occupation. In addition, labor and capital affect urban economic efficiency and competitiveness [51].

Output consists of desirable output and undesirable output. Economic output represents the desirable output, as measured by the added value of secondary and tertiary industries. The total urban industrial wastewater emissions, industrial exhaust gas emissions, smoke, and dust emissions reflect the undesirable output.

#### 3.2.2. Control Variables

The selection of control variables is as follows: (1) Urban population (POP). The increase in urban population can bring economies of scale to urban development, thereby influencing the ULUE [52]. (2) The ratio of the output value of the tertiary industry to the secondary industry is used to estimate the industrial structure (IS). The larger the ratio, the more optimized the industrial structure and the higher the ULUE [53]. (3) The per capita road area (PROAD) reflects the convenience of the city. The more convenient the traffic, the higher the accessibility and the faster the economic flow. (4) The proportion of fiscal expenditure (FE) reflects the degree of local government intervention. Local governments in China generally use administrative measures to intervene in land use and to increase land taxation, which will affect the economic output of land use [54]. (5) GDP per capita (PGDP) represents the level of economic development. The higher the per capita GDP, the more reasonable the allocation of resources. (6) Per capita foreign direct investment (PFDI). With the deepening of globalization, foreign capital has transformed global forces into localized forces through location selection, which has a significant impact on urban scale economic and technological progress, thereby changing the ULUE [21]. (7) Per capita construction land area (PCL)indicates the carrying capacity and the development intensity of urban land. (8) The proportion of real estate investment (PRES) is used to indicate land marketization. The land market reflects the allocation of land resources and the behavior of land users by changing the mechanisms of supply and demand, price, and competition [10]. (9) The proportion of spending on science and technology (PTE) indicates the level of technological innovation. (10) The comprehensive utilization rate of industrial solid wastes (CUS) represents urban environmental regulations, which influence the output of ULUE. Table 2 provides the descriptive statistics of all variables.

### 3.3. Model Selection

#### 3.3.1. Difference-in-Differences Model (DID)

Seeing as how the difference-in-differences (DID) model could effectively capture the net effect of the HSR implementation policy [47], this paper used the DID model to investigate the impact of HSR on the change in ULUE. The prefecture-level cities that have opened HSR between 2003 and 2018 were treated as the experimental group and cities without HSR were treated as the control group. Thus, the total pool of 4544 sample data was obtained. The model is as follows:(1)$$Y_{it}=\alpha_{0}+\alpha_{1}*Treated_{i}+\alpha_{2}*Period_{t}+\alpha_{3}Treated_{i}*Period_{t}+\varepsilon_{it}$$
where $Y_{it}$ represents the ULUE of city i in year t and $Treated_{i}*Period_{t}$ represents a binary dummy variable. If cities with HSR in year t, the $Treated_{i}*Period_{t}=1$; otherwise $Treated_{i}*Period_{t}=0$. $\alpha_{0}$, $\alpha_{1}$, $\alpha_{2}$ and $\alpha_{3}$ represent constants (the regression coefficients of grouping variables, time variables, and the net effect of the opening of HSR on ULUE, respectively). If the $\alpha_{3}$ is statistically significant and positive, it indicates that the opening of HSR can significantly improve the ULUE.

Given the progressive construction of a HSR network, a multi-stage difference-in-differences model (multi-stage DID model) was utilized to analyze the dynamic effects of HSR gradually opening. The multi-period DID model is as follows:(2)$$\ln Y_{it}=\beta_{0}+\beta_{1}Wstation_{it}+\gamma Controls_{it}+\mu_{i}+f_{t}+\varepsilon_{it}$$
where $Wstation_{it}$ indicates whether the city i opened HSR in year t. If cities opened the HSR, the $Wstation_{it}=1$; otherwise, $Wstation_{it}=0$. Due to the inability to unify the time for the construction of HSR in cities at various levels $\left(Period_{t}\right)$, $\alpha_{1}$ and $\alpha_{2}$ will no longer exist. At the same time, the fixed effects of region and time are adopted in the new model.$\sum_{n}Controls_{it}$, $\beta_{0}$, and γ are control variables, constants, and regression coefficients of control variables, respectively. $\beta_{1}$ represents the degree of impact of the opening of HSR on the ULUE. If, $\beta_{1}>0$, it means that the opening of HSR can promote the ULUE; otherwise, it will decrease the ULUE.

Further, this paper replaces $Wstation_{it}$ with $Station_{it}$ and $Route_{it}$ to test the impact of the number of HSR stations and routes on the ULUE. $Station_{it}$ and $Route_{it}$ represent the cumulative number of HSR stations or routes opened of city i in year t. The final model is as follows:(3)$$\ln Y_{it}=\alpha_{0}+\alpha_{1}Station_{it}+\gamma Controls_{it}+\mu_{i}+\delta_{i}+\varepsilon_{it}$$
(4)$$\ln Y_{it}=\alpha_{0}+\alpha_{1}Route_{it}+\gamma Controls_{it}+\mu_{i}+\delta_{i}+\varepsilon_{it}$$

#### 3.3.2. Threshold Model

To examine whether the number of HSR stations and routes have any threshold effect on ULUE, regression analyses were performed based on the samples before and after the threshold, respectively. The model used is as follows:(5)$$Y_{it}=\mu_{i}+\theta_{1}X_{it}I(q_{it}\leq \gamma )+\theta_{2}X_{it}I(q_{it}>\gamma )+\ell_{it}$$
where $Y_{it}$, $X_{it}$, $q_{it}$, γ and I represent the dependent variable, explanatory variable, threshold variable, threshold value, and characteristic function, respectively. If I=1, it means that the function meets the conditions shown in the brackets; otherwise, I=0. $\ell_{it}$ represents random interference.

There are three issues to be solved in the threshold model: determining the threshold, testing the importance and authenticity of the threshold, and discriminating the number of thresholds. The threshold is modelled as follows:(6)$$\hat{\gamma }=\underset{\gamma }{arc\min}\;S_{1}(\gamma ),\gamma \in \eta$$

Then, there is a significance test of the threshold effect. If the null hypothesis test passes, there is no threshold effect; otherwise, the threshold effect will be determined [34]. Specifically, The Lagrangian Multiplier Test is used as follows:(7)LR(γ)=S1(γ)−S1(γ^)σ^2 where $S_{1}(\gamma )$ and $S_{1}(\hat{\gamma })$ represent the residual sum of squares without threshold and under the threshold effect, respectively.

Finally, the above model can be changed as follows, and the threshold effect of the number of HSR route is consistent with the following calculation method.
(8)$$\begin{array}{l} \ln Y_{it}=\alpha_{0}+\alpha_{1}Station_{\;\;\;it}(Popd_{\;\;\;it}\leq \gamma_{1})+\alpha_{2}Station_{\;\;\;it}(Popd_{\;\;\;it}>\gamma_{1})+\gamma Controls_{\;\;\;it}\\ \;+\mu_{i}+\delta_{i}+\varepsilon_{it} \end{array}$$
(9)$$\begin{array}{c} \ln Y_{it}=\alpha_{0}+\alpha_{1}Station_{it}(Popd_{it}\leq \gamma_{1})+\alpha_{2}Station_{it}(Popd_{it}>\gamma_{1})+\alpha_{3}Station_{it}\\ (\gamma_{1}<Popd_{it}\leq \gamma_{2})+\alpha_{4}Station_{it}(Popd_{it}>\gamma_{2})+\gamma Controls_{it}+\mu_{i}+\delta_{i}+\varepsilon_{it} \end{array}$$

The above two equations represent the single threshold model and the dual threshold model, respectively. The selection of the model is determined by the likelihood ratio test of the threshold variable.

## 4. Results

### 4.1. Annual Changes in ULUE

Figure 2 shows the changes in ULUE of the cities with HSR and without HSR from 2003 to 2018. The gap in ULUE between the cities with HSR and without HSR is highly significant. In order to examine whether there is a statistically significant causal relationship between HSR and ULUE, an econometric model was used for further analysis.

### 4.2. Results of DID Regression

#### 4.2.1. Baseline Regression

First, a multi-phase DID model was utilized to examine the impact of HSR on ULUE (based on Equations (2)–(4)). Considering the existence of heterogeneity of cities in terms of urban population density, locational and temporal variations, this paper controls the individual effect and time effect.

Table 3 shows the baseline regression results of the DID model. The coefficients of Wstation in Model 1, Station in Model 3, and Route in Model 5 are positive, respectively, with *p* < 0.01 or *p* < 0.05. The results indicate that the opening of HSR and the adding of a HSR station and route are associated with an increase in ULUE in a city by 0.021, 0.004, and 0.013 compared to cities without HSR, respectively. This is due to the increasing of market accessibility of cities along the railway [48], the promotion of industrial agglomeration and the growth of employment [49], and the improvement of urban productivity [50]. These findings support that ULUE can gradually enhance as large-scale HSR construction is ongoing.

Further, control variables were added to the models. The coefficients of Wstation in Model 2, Station in model 4, and Route in Model 6 have decreased as compared to those in Models 1, 3, and 5. This indicates that the HSR has effectively controlled the interference of other factors. Finally, this paper controls the variable of Station (Model 7). The coefficient of Route in Model 7 is positive (with *p* < 0.1), indicating that adding a HSR route will increase the ULUE by 0.012 in the case that the city has opened HSR. These findings support the above theoretical analysis. HSR can effectively promote the ULUE, and the HSR stations often become new hotspots for urban activities [51], resulting in increased scale efficiency and technological efficiency.

This paper further estimated the effect of other factors. The coefficients of lnPOP, lnPROAD, lnPGDP, lnPFDI and lnCUS in Models 2, 4, and 6 are positive, with *p* < 0.01. Those indicate that urban population, road area per capita, GDP per capita, per capita foreign direct investment, and comprehensive utilization rate of solid wastes would cause an increase in ULUE. The coefficients of lnFE in Models 2, 4, and 6 are negative (with *p* < 0.001), which indicates that the ratio of local fiscal expenditure to GDP will cause the ULUE to decline. The possible reason for this is that the low-cost supply of industrial land has led to the extensive and inefficient use of urban land [10].

#### 4.2.2. Regional Heterogeneity

As mentioned above, economic development and urban growth in China exhibit clear spatial disparities, increasing from the poorer Western region to the richer Eastern region. This paper future tests whether HSR has a regional heterogeneity regarding ULUE in Chinese cities. According to the level of economic development, 284 cities were divided into three regions: the East (Earea), the central (Carea) and West (Warea), respectively. The interaction term variables of city location and HSR were added to equation (2). Similarly, time and individual variables were controlled.

Table 4 shows the results of heterogeneity. As shown in Table 4, the coefficients of Wstation × Earea in Model 8 and Wstation × Warea in Model 11 are positive (with *p* < 0.05), which indicates that the first opening HSR can significantly increase the ULUE of the East and the West. However, the coefficients of Wstation × Carea in Model 14 are insignificant, indicating that the opening of the HSR for the first time has no significant impact on ULUE in the central regions.

The coefficients of Station × Earea, Station × Warea, and Station × Carea in Models 9, 12, and 15 in Table 4 show a significant positive impact on the ULUE. In addition, the coefficients of HSR stations exhibit clear spatial disparities, descending from the West to the East. Therefore, there is reason to believe that the construction of an HSR station within a certain period of time can positively affect scale efficiency in the underdeveloped regions of the central and western regions. In addition, the East, with good levels of economic development and urbanization, should pay attention to waste discharge supervision and domestic waste treatment while planning land use. These finding capture the fact that increasing the number of HSR stations will trigger the growth of desired output and improve the ULUE.

The coefficients of Route × Earea, Route × Warea and Route × Carea in Model 10, 13 and 16 shows that the coefficient of Route × Earea is not significant, implying that the number of HSR routes cannot effectively promote efficiency, considering that cities in the East are relatively developed. Meanwhile, the coefficients of Route × Warea and Route × Carea are significantly positive, which points to the fact that the number of HSR routes in developing cities will improve urban development, thereby increasing the ULUE. As cities continue development and become mature, the number of HSR routes may have no effective effect of the ULUE considering the upward trend of land-use costs. For megacities, the further improvement of ULUE requires a series of structural changes, due to the fact that each megacity’s own ULUE has reached a relatively high level.

The above research results convey a phenomenon that the HSR has various effects on ULUE in different regions. It has a higher effect on the ULUE in the central and Western regions than it does in the Eastern region. First, the extensive technology dissemination space and fast technology update speed in the eastern region have weakened the impact of the technological efficiency brought by HSR on ULUE. Second, the space—time compression effect brought about by HSR and the change in urban accessibility have further enhanced the attractiveness of the Eastern region to high-quality population and innovative enterprises. As a result, the resources of the central and Western regions flow to the Eastern regions, eventually leading to an increase in urban land demand in Eastern cities but also causing a mirrored decrease in central and Western cities. Hence, the scale efficiency and the complementary role of technological innovation are more prominent in the central and Western regions. Finally, the ULUE of cities in the central and Western regions is lower than that in the Eastern region, which leads to greater room for improvement.

### 4.3. Threshold Regression

Given that the HSR directly serves the population, the urban population density is used as a threshold variable according to the relevant research of Li (2020) [15]. Table 5 reports the results of the threshold effect. The results of Models 17–18 show that a single threshold regression model is more appropriate to be applied to the impact of the number of HSR stations and routes on ULUE.

The coefficient of lnStation in Model 17 shows that when urban population density is less than the threshold value of 794,889 people/km^2^, ULUE is associated with a 0.022 increase, given the influence of the number of HSR stations. Otherwise, the ULUE would associate with a 0.043 increase. When the urban population density of a city with an HSR station exceeds 794,889 persons/km^2^, the positive impact of the number of HSR stations on the ULUE is likely to be two times larger than urban population density is less than 794,889 persons/km^2^_._

Similarly, the coefficient of lnRoutein Model 18 shows that when urban population density is less than the threshold value of 1,002,668 people/km^2^, ULUE is associated with a 0.127 increase, given the influence of the number of HSR routes. Otherwise, the ULUE would associate with a 0.025 increase. Thus, the higher the population density of the city, the more obvious the efficiency brought by the HSR.

### 4.4. Robustness Test

The placebo test was used to examine the reliability of regression analysis. The most important and key prerequisite for DID is that the treatment group and the control group must have the same development trend before the policy is implemented. At the same time, there will be other policies that may affect trend changes. To solve this problem, a placebo test was used.

The specific operation assumes that the opening time of the HSR was 5 years earlier than reality for verification. A “pseudo-HSR-opening” variable (HSRT_before5) was generated, which can test the effect of treatment. If the research results were still consistent with the original analysis, that indicates that there are other factors that may affect the ULUE. Otherwise, the baseline results are considered reasonable.

Table 6 provides the results of the placebo test. The coefficients of in Models 19–22 are not statistically important, which indicates that the impact observed in Table 3 does reflect the impact of the opening of the HSR, rather than other unobservable factors. Therefore, the DID method is sufficient in this paper.

## 5. Discussions

China’s current planning practice emphasizes the improvement of ULUE within the scope of urban internal resource and environmental carrying capacity [55]. However, the urban environment is an open system and non-local connections play a key role in resource reduction and environmental conservation. Hence, this paper can provide some insights into improvement of ULUE for responding to sustainable development.

First, in order to improve the ULUE, it is necessary to carefully implement land use policies related to the development of HSR based on the specific conditions of different cities. For the central and Western regions in China, the opening of a HSR and the numbers of HSR stations and routes can greatly increase the ULUE [56]. Thus, it is necessary to formulate corresponding policies to promote the adoption of HSR, which will make a greater contribution to improving the ULUE of cities in the stage of accelerated economic development [13]. One should note that this discovery may also help some developing countries (such as India, Indonesia, and Malaysia) where the development of HSR is ongoing or being debated [57]. Given the rising cost of land and the higher ULUE in the Eastern region in China, the further improvement of ULUE requires improvement in the land use structure [34]. Similarly, this discovery may also help policymakers in other countries with higher levels of economic development to better understand the opportunities of HSR.

Second, attention should be paid to issues such as population loss in the midwest and smaller cities. Compared with the cities in the East, the ULUE of midwestern cities are more vulnerable to the impact of HSR. The agglomeration of factors flowing to the Eastern region leads to an increase in urban land demand in Eastern cities but a decrease in central and Western cities [58]. The government in the areas where the HSR is opened should provide more specific policy measures to prevent the outflow of less capital and talents, such as tax cuts and housing subsidies for talented labor [49]. Further, the positive effect of HSR stations and routes on ULUE are more significant in cities with a relatively high urban population density. Therefore, more specific HSR planning with population scale and regional development differences in mind must be implemented to maximize the economic benefits of the system [15].

## 6. Conclusions

Improving the urban land use efficiency is the only way to keep urban growth under limited land supply and resource constraints. The impact of social and economic factors has been widely discussed, but the impact of high-speed rail is rarely considered. China has the world’s largest high-speed rail network, and its impact on urban land use efficiency cannot be ignored. In order to enrich the discussion between them, this paper investigates the impact of high-speed rail on urban land use efficiency. The results show that there is a 0.021 increase in the urban land use efficiency after opening the HSR. The number of HSR stations and routes can increase the urban land use efficiency by 0.004 and 0.013, respectively. In addition, this positive promotion effect is more obvious in the midwestern cities and cities with a greater urban population density. Specifically, the positive impact of the number of HSR stations on the urban land use efficiency in cities with an urban population density exceeding 795 persons/km^2^ is two times larger than that in cities with an urban population density of less than 795 persons/km^2^.

The possible implications for practice are threefold. First, from the perspective of long-term urban development, the high-speed rail network connecting the developed cities in the East and the underdeveloped cities in the midwestern regions can effectively reduce the difference in accessibility and should be actively encouraged. High-speed rail strengthens non-local connections between cities, breaks the constraints of administrative boundaries on the circulation of elements, and promotes structural changes in the urban economy.

Second, in China’s future urban planning, the government should give proper preference to underdeveloped regions in order to achieve the most effective use of resources and the most balanced development of different regions. The midwestern regions urgently need the frequent flow of production factors such as personnel and capital to achieve rapid development, which requires the full use of the advantages of high-speed rail network.

In addition, all cities within the high-speed rail network can benefit from interconnection. China’s current planning practice emphasizes controlling the scale of cities to deal with the problem of land scarcity, but this idea of limiting capacity is static and can easily cause distortions in the allocation of land resources. Although land is a non-tradable resource, the fact is that high-speed rail has built a broader and freer product market by strengthening the circulation of factors, which will further change the pattern and intensity of urban land use. Therefore, in this regard, strengthening the construction of high-speed rail to integrate the urban network can be an effective measure to improve land use efficiency and achieve sustainable development.

In addition, there are several areas for further research. First, future research needs to consider the fact that non-HSR cities may suffer from the spillover effects of HSR cities. Second, future research needs to refine the adjustment of administrative divisions in different years. Finally, future research can investigate the influence of other factors of threshold effects, such as the connectivity of the HSR network and the purpose of HSR travel.

## Figures and Tables

**Figure 1 ijerph-18-10043-f001:**
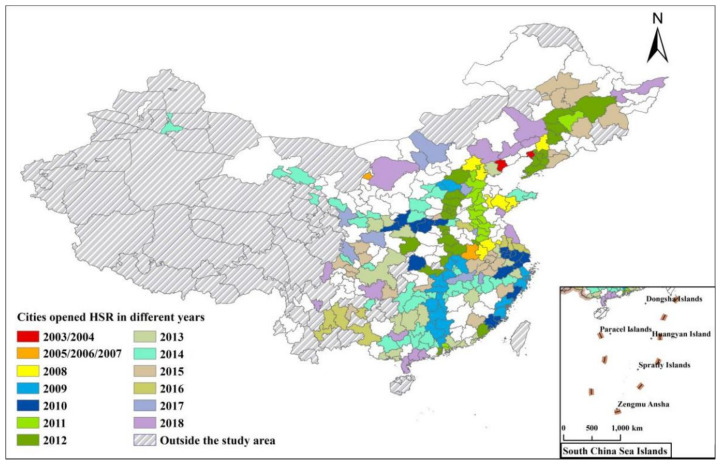
Cities that opened HSR in China from 2003 to 2018.

**Figure 2 ijerph-18-10043-f002:**
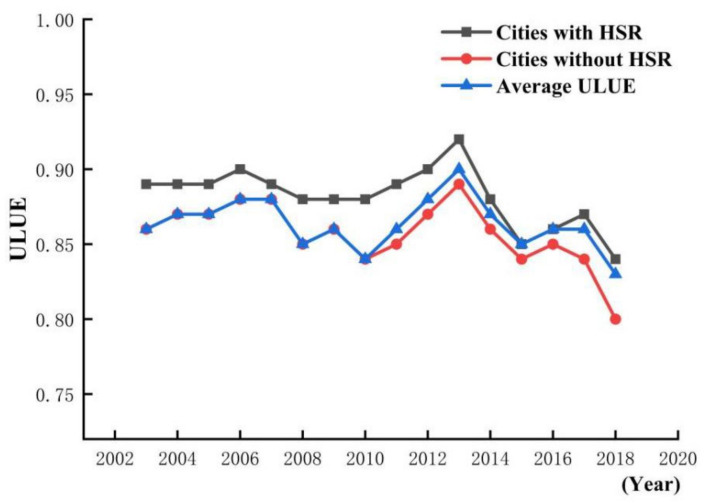
The changes in ULUE of the cities with HSR and without HSR.

**Table 1 ijerph-18-10043-t001:** Input—output indicators for measuring ULUE.

Index Type	First-Class Indicators	Second-Class Indicators
Input	Land	Urban land (km^2^)
	Labor	Numbers of employees in a secondary and tertiary industry (10 thousand people)
	Capital	Investment in fixed assets (100 million yuan)
Output	Desirable output	Added value of the secondary and tertiary industries (100 million yuan)
		Area of green space (km^2^)
	Undesirable output	Urban industrial SO_2_ emission (ton)
		Urban industrial smoke/dust emission (ton)
		Urban land (km^2^)

**Table 2 ijerph-18-10043-t002:** Descriptive statistics.

Variables		Meaning	Number of Samples	Mean	Standard Deviation	Min	Max
High-speed rail variables	Wstation	Whether to open HSR	4544	0.268	0.443	0.000	1.000
	Station	Number of HSR station	4544	0.740	1.642	0.000	14.000
	Route	Number of HSR route	4544	0.362	0.689	0.000	5.000
Explained variables	lnULUE	Urban land use efficiency	4544	−0.162	0.216	−2.511	0.000
Control variables	lnPOP	Urban population	4544	5.867	0.685	2.795	8.133
	lnIS	Industrial structure	4544	−0.234	0.452	−2.361	4.403
	lnPROAD	Road area per capita	4544	0.906	0.968	−2.487	4.291
	lnFE	Proportion of fiscal expenditure	4544	−1.993	0.522	−6.185	0.955
	lnPGDP	Per GDP	4544	10.153	0.904	7.545	13.185
	lnPFDI	Per capita foreign direct investment	4544	3.751	1.808	−2.143	9.613
	lnPCL	Per capita construction land area	4544	−8.225	0.424	−9.311	−5.451
	lnPTE	Proportion of spending on science and technology	4544	−6.408	1.323	−11.457	−1.219
	lnPRES	Proportion of real estate investment	4544	−2.109	0.771	−5.09	1.729
	ln CUS	Comprehensive utilization rate of solid wastes	4544	4.285	0.487	−1.427	11.358

**Table 3 ijerph-18-10043-t003:** Baseline regression results of the DID model.

Variables	1	2	3	4	5	6	7
Wstation	0.021 ** (2.72)	0.017 *** (2.21)					
Station			0.004 ** (2.030)	0.003 *** (5.420)			0.001 (−0.090)
Route					0.013 *** (2.650)	0.011 ** (2.190)	0.012 * (1.710)
lnPOP		0.043 *** (4.270)		0.050 *** (4.170)		0.057 *** (5.320)	
lnIS		0.010 (0.710)		0.007 (0.680)		0.007 (0.710)	
lnPROAD		0.007 *** (6.830)		0.049 *** (6.850)		0.049 *** (6.830)	
lnFE		−0.010 *** (2.730)		−0.027 *** (2.750)		−0.027 *** (2.780)	
lnPGDP		0.016 *** (3.550)		0.059 *** (3.600)		0.059 *** (3.630)	
lnPFDI		0.005 *** (−2.750)		−0.013 *** (−2.720)		−0.013 *** (−2.670)	
lnPCL		0.019 (0.210)		0.004 (0.220)		0.004 (0.210)	
lnPTE		0.006 (−0.960)		0.006 (−1.030)		0.005 (−0.920)	
lnPRES		0.004 (1.630)		0.007 (1.640)		0.007 (1.640)	
lnCUS		0.007 *** (2.730)		0.019 *** (2.710)		0.019 *** (2.710)	
Fixed	Yes	Yes	Yes	Yes	Yes	Yes	Yes
Obs	4544	4544	4544	4544	4544	4544	4544
R-squared	0.755	0.751	0.685	0.792	0.762	0.747	0.825
F-test	5215.028	5293.087	5211.527	5290.985	5214.635	5292.975	5212.643

Notes: *t* statistics in parentheses. * *p* < 0.10, ** *p* < 0.05, *** *p* < 0.01.

**Table 4 ijerph-18-10043-t004:** The test for regional heterogeneity.

Variables	8	9	10	11	12	13	14	15	16
$\begin{array}{c} Wstation\times \\ Earea \end{array}$	0.039 ** (2.75)								
$\begin{array}{c} Station\times \\ Earea \end{array}$		0.013 * (1.830)							
$\begin{array}{c} Route\times \\ Earea \end{array}$			0.008 (1.020)						
$\begin{array}{c} Wstation\times \\ Warea \end{array}$				0.160 ** (3.120)					
$\begin{array}{c} Station\times \\ Warea \end{array}$					0.043 * (1.850)				
$\begin{array}{c} Route\times \\ Warea \end{array}$						0.030 * (1.69)			
$\begin{array}{c} Wstation\times \\ Carea \end{array}$							0.017 (0.870)		
$\begin{array}{c} Station\times \\ Carea \end{array}$								0.021 *** (3.352)	
$\begin{array}{c} Route\times \\ Carea \end{array}$									0.028 *** (4.85)
Controls	Yes	Yes	Yes	Yes	Yes	Yes	Yes	Yes	Yes
Fixed	Yes	Yes	Yes	Yes	Yes	Yes	Yes	Yes	Yes
Obs	1794	1794	1794	992	992	992	1758	1758	1758
R−squared	0.871	0.810	0.886	0.719	0.725	0.772	0.824	0.838	0.848
F−test	710.253	721.351	715.235	253.652	254.378	265.321	685.986	691.249	687.921

Notes: *t* statistics in parentheses. * *p* < 0.10, ** *p* < 0.05, *** *p* < 0.01.

**Table 5 ijerph-18-10043-t005:** Regression results of the threshold effect.

Variables	17	Variables	18
	0.022 ** (3.125)		0.127 ** (2.151)
	0.043 *** (8.124)		0.025 *** (6.125)
Constant	−0.184 *** (−22.110)	Constant	−1.019 *** (−3.510)
Obs	4544	Obs	4544
R-squared	0.911	R-squared	0.905
Threshold	=794.889	Threshold	=1002.668

Notes: *t* statistics in parentheses. ** *p* < 0.05, *** *p* < 0.01.

**Table 6 ijerph-18-10043-t006:** Placebo test results of the DID model.

Variables	19	20	21	22
Area	Whole area	East	Central	West
HSRT_before5	0.009 (1.15)	0.009 (1.35)	0.035 (2.18)	0.035 (1.35)
Constant	0.249 (8.02)	0.110 * (5.14)	−0.085 (−1.78)	0.717 (2.42)
Fixed	Yes	Yes	Yes	Yes
Obs	3124	1232	1208	682
R-square	0.863	0.983	0.976	0.877
F-test	139.21	379.32	289.86	42.60

Note: *t* statistics in parentheses. * *p* < 0.10.

## Data Availability

The data presented in this study are available on request from the author. The data are not publicly available due to privacy. Images employed for the study will be available online for readers.

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
