# Peer review of "Change in Urban Land Use Efficiency in China: Does the High-Speed Rail Make a Difference?"

_ijerph, 2021, doi:10.3390/ijerph181910043_

Round 1
Reviewer 1 Report
The conclusions should be more profound and detailed. In my opinion, the changes introduced in this section are not enough. After this improvement, the paper submission can be reconsidered for publication. However, in my opinion the current version of this manuscript section was not sufficiently improve. Thank you!
Reviewer 2 Report
I find that the authors have put considerable effort in revising and improving the manuscript according to the review comments. The manuscript is very much improved and I have no problem in recommending it for publication.
Author Response
Thank you for your valuable comments and patience review.
Round 2
Reviewer 1 Report
I have no additional comments. Please, check style and minor English typos.
This manuscript is a resubmission of an earlier submission. The following is a list of the peer review reports and author responses from that submission.
Round 1
Reviewer 1 Report
Thank you for answering the comments. However, new comments based on previous ones must be highlighted a newly explained and improved:
a. OK.
b. Novelty is not currently included in the revised version of the manuscript. A paragraph including this issue must be introduced in the revised version.
c. I could not be identified the word "aim" or "objective" in the revised version of the manuscript. The main objective of the paper has to be explicitly explain on it.
d. Ok.
e. Perfect. Ok.
f. Ok.
g. OK.
h. That's right.
i. Discussion section has been improved in the revised version of the manuscript. However, no references give support to the assumptions stated on this particular section. Please, author must justify their discussion based on previous research work developed in the area.
j. Ok.
k. Future research work must be included in conclusion section, not in discussion section.
l. Conclusions are summarised in a really naif paragraph. This section must be completely rewritten and restructured.
m. ok.
n. Which figure is new?
o. Reconsideration after new major revision. In my opinion, the current revision of the manuscript was not enough.
Reviewer 2 Report
This manuscript takes 284 prefecture-level cities and county-level cities as the research subject, and use a multi-period Difference-in-Difference model and a threshold model to examine the impact of HSR on urban land use efficiency with urban population, GDP, per capita road area, and the proportion of public investment as control variables. On this basis, the manuscript explores the regional heterogeneity. The article has a clear logical framework and rich contents. It has certain research significance with respect to the HSR-related research fields, but there are some problems that undermine the integrity of the manuscript, as well as the overall narrative. They are as follows:
- The authors mentioned that raising the urban land use efficiency is the straightforward idea to meet the increasing urban growth and limited supply of urban land, and began to discuss the impact of HSR on urban land use efficiency directly. Given that there may be many factors that affect the efficiency of land use, why in-depth discussions on the impact of HSR are important? I recommend the authors elaborate more on any relevant background knowledge here.
- The purpose and significance of the research are not clear. What is the frontier of the impact of HSR on urban land efficiency, and what new ideas has this research proposed in terms of research topics and methods? What are the academic value and practical significance of the research topic? The introduction does not systematically elaborate on these points. I recommend the authors specifically state the knowledge gaps, research questions that address them, and the respective hypotheses, as well as this research’s innovations and their significance.
- Regarding the insufficient review of the role of HSR on the ULUE, the authors mainly focused on the research of Chinese scholars, and foreign literature was not sorted out enough. It is recommended to discuss this issue from multiple perspectives, such as research perspectives, research content, and research methods. In addition to Chinese journals, please look for major international sources such as Journal of Transport Geography, Transportation Research Parts A, B, and D, etc. Dissertations are also important sources of references, such as Wanli Fang’s PhD Dissertation at MIT (2013), which addressed a very similar question.
- The choice of model variables appears arbitrary. The authors chose variables mainly referring to several existing research results. It is necessary to control the impact factors other than high-speed rail, such as population density, construction land area per capita, road area per capita, investment in science and technology, comprehensive utilization rate of industrial solid wastes, etc. Otherwise, one can be hardly convinced that the findings are valid.
- In analyzing the land use efficiency, the authors used all types of urban land, which actually include the land occupied by the HSR itself. Since the HSR system is land-consuming and occupies a non-negligible part of land, it would be problematic for not ruling out this part of land for the purpose of analyzing the “impact” of HSR, which should emphasize on HSR’s INDUCED effects on land use efficiency, rather than the impact of HSR’s own land occupation.
- The authors need to elaborate on the possible underlying causes for the results. For example, what are the possible reasons for the finding “compared with the cities in east, the urban land use efficiency of mid-western cities is more vulnerable to the impact of HSR.”